# Black holes, quantum chaos, and the Riemann hypothesis

**Panagiotis Betzios[1], Nava Gaddam[2⋆] and Olga Papadoulaki[3]**

**1** Crete Center for Theoretical Physics, Institute for Theoretical and Computational Physics, Department of Physics, University of Crete 71003, Heraklion, Greece.
**2** Institute for Theoretical Physics and Center for Extreme Matter and Emergent Phenomena, Utrecht University, 3508 TD Utrecht, The Netherlands.
**3** International Centre for Theoretical Physics, Strada Costiera 11, 34151 Trieste, Italy.

⋆ gaddam@uu.nl

## Abstract

Quantum gravity is expected to gauge all global symmetries of effective theories, in the ultraviolet. Inspired by this expectation, we explore the consequences of gauging CPT as a quantum boundary condition in phase space. We find that it provides for a natural semiclassical regularisation and discretisation of the continuous spectrum of a quantum Hamiltonian related to the Dilation operator. We observe that the said spectrum is in correspondence with the zeros of the Riemann zeta and Dirichlet beta functions. Following ideas of Berry and Keating, this may help the pursuit of the Riemann hypothesis. It strengthens the proposal that this quantum Hamiltonian captures the near horizon dynamics of the scattering matrix of the Schwarzschild black hole, given the rich chaotic spectrum upon discretisation. It also explains why the spectrum appears to be erratic despite the unitarity of the scattering matrix.

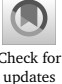

# 1 Introduction

It has long been proposed that quantum gravity has no global symmetries; all global symmetries of effective theories shall be gauged [1–3]. This expectation also applies to discrete global spacetime symmetries [4,5]. In fact, there is an old idea that shows how discrete global symmetries may remain in the infrared, upon Higgs-ing local gauge symmetries in the ultraviolet [6]. This article concerns a global discrete spacetime symmetry, namely CPT. Indeed, in quantum gravity, a definition of global CPT requires the specification of a choice of time, a choice of gauge. It appears to intrinsically be an effective symmetry. For instance, should there be a symmetric phase of quantum gravity, a possible symmetry breaking mechanism could lead to a vacuum expectation value for the metric [7], and consequently an effective global discrete CPT symmetry in the low energy semiclassical description. The larger gauge symmetry group might then subsume such a discrete CPT symmetry.

Often the ultraviolet physics is attributed to new degrees of freedom (such as strings). Could it be that intricate microscopic gauge symmetries and dynamics result in an effective boundary condition for the low energy description? There is a precedent for such a mechanism in the Callan-Rubakov effect. A possible relevance to black hole physics was articulated in [8] and a similar effect could be at play in Euclidean quantum gravity [9]. Lacking an exact understanding of the microscopic theory, in this article, we will assume as a postulate that CPT is gauged and analyse its consequences in an example motivated by the study of quantum black holes.

In a seemingly unrelated pursuit, Berry and Keating proposed [10,11] that a rich quantum spectrum could be generated for the Dilation operator, by applying discrete phase space boundary conditions. This was inspired by a conjecture due to Hilbert and Polya.[1] The latter is the statement that the zeros of the Riemann zeta function arise as the spectrum of a self-adjoint operator with real eigenvalues. The spectrum of the Riemann zeros is very rich, a smooth approximation of which is known to produce the statistics of the Gaussian Unitary Ensemble of random matrices [13]. Nevertheless, several ambiguities remain to date: the appropriate self-adjoint operator, the exact boundary condition to impose that determines the space of functions upon which the said operator acts, and the physical origin for this choice. Needless to say, there have been several proposals for such an operator; see [14] and references therein.

Could it then be that the effect of the proposed gauging of CPT symmetry provides for a quantisation of semiclassical continuous spectra? Drawing on work towards constructing a unitary S-matrix for a Schwarzschild black hole [15–17], we explore the consequence of such a gauging as a quantum boundary condition in phase space. First, we find that the spectrum of the Dilation operator is naturally discretised and is given by the zeros of Riemann zeta and Dirichlet beta functions. Following the observation in [17] that such a Dilation operator captures the near horizon dynamics of the black hole scattering matrix, we argue that the boundary condition results in a discrete spectrum of black hole microstates. Despite the seemingly simple classical Hamiltonian, an almost erratic spectrum can arise. This is in accord with the expectation that black holes have a "fast scrambling", quantum chaotic nature [18–22]

# 2 The Dilation operator and symmetries

In [17], it was shown that two different Hamiltonians with their associated phase space structure, result in exactly the same scattering matrix. One is a collection of a tower of independent

---

[1]The first documented appearance of which is in [12].

inverted harmonic oscillators labelled by spherical harmonic indices $l, m$:

$$\hat{H} \;=\; \frac{1}{2} \sum_{l\,m} \left(\hat{U}_{lm}\hat{V}_{lm} + \hat{V}_{lm}\hat{U}_{lm}\right), \tag{1}$$

$$\left[\hat{V}_{lm}, \hat{U}_{l'm'}\right] = i\hbar\lambda_l \delta_{ll'}\delta_{mm'}, \quad \lambda_l = \frac{8\pi G}{R^2(l^2+l+1)},$$

where the operators $\hat{U}$ and $\hat{V}$ define the light-cone positions of the partial waves. The parameter $G/R^2$ is dimensionless and will be interpreted shortly. In this description, an elementary cell in phase space, and the commutation relations, depend on the partial wave number $l$. The second description has a common phase space cell size, and therefore a single unit of time for the entire tower:

$$\hat{H} \;=\; \frac{1}{2} \sum_{l\,m} \left(\hat{U}_{lm}\hat{V}_{lm} + \hat{V}_{lm}\hat{U}_{lm}\right) + \frac{R^2(l^2+l+1)}{8\pi G}, \quad \left[\hat{V}_{lm}, \hat{U}_{l'm'}\right] \;=\; i\hbar\delta_{ll'}\delta_{mm'}. \tag{2}$$

The lightcone position operator in both descriptions may be expanded in partial waves as follows

$$\hat{U}(\Omega) \;=\; \sum_{lm} \hat{U}_{lm} Y_{lm}(\Omega).$$

Both these Hamiltonians yield the same scattering matrix [17]. Moreover, both of them are clearly diagonal in the partial wave basis. The complete wavefunctions are specified in the lightcone position basis using the two sphere coordinate, say $\Omega$, by

$$\langle U;\Omega|\Psi\rangle \;=:\; \Psi(U;\Omega) \;=\; \sum_{l\,m} \Psi_{lm}(U) Y_{lm}(\Omega), \tag{3}$$

$$\langle U;l\,m|\Psi\rangle \;=:\; \Psi_{lm}(U).$$

One can also describe the wavefunctions the $V$-basis i.e. $\langle V;l\,m|\Psi\rangle \;=:\; \Psi_{lm}(V)$. The $V$-basis naturally describes the ingoing modes, while the outgoing modes are naturally described in the $U$-basis. Since there is no spin dependence in the scattered states and their associated dynamics, the index $m$ represents a $(2l+1)$-fold degeneracy in the spectrum.

**Connection with the Schwarzschild black hole** The scattering matrix arising from the two Hamiltonians described above is identical to the one obtained from the gravitational (shock-wave) back-reaction computation in [15, 16], as was shown in [17]. For this comparison, the constant $G$ is identified with Newton's constant and $R = 2GM$, with the radius of a Schwarzschild black hole with mass $M$. That the scattering is diagonal in the partial wave basis is owed to the spherical symmetry of the black hole.

The wavefunctions of eqn. (3) describe partial waves in the Kruskal coordinates of an eternal Schwarzschild black hole, which carries no charge other than its mass $M$. The metric takes the form

$$ds^2 \;=\; -\frac{32G^3M^3 e^{-c^2r/2GM}}{c^6 r}\, dU\, dV + r^2 d\Omega^2, \tag{4}$$

$$UV \;=\; \left(1 - \frac{c^2 r}{2GM}\right) e^{c^2 r/2GM}.$$

The Hamiltonians (1) and (2), generate the dynamical evolution of the partial waves on this background. Furthermore, their evolution is according to the clock of a boosted observer that remains at a distance from the black hole, reaching asymptotic null infinity [17]. The energy then is measured in such an observer's natural units [23] as $E/\hbar = \omega/\kappa = 4GM\omega/c^2$, where

$M$ is the mass of the said black hole, $\kappa$ is the surface gravity, and $\omega$ is the energy (in SI units) that the asymptotic observer registers. Henceforth, we set $c = 1$ for simplicity. The coordinate transformation from Kruskal coordinates to those of the boosted observer is

$$U = -e^{-u/4GM}, \quad V = e^{v/4GM}, \tag{5}$$

in terms of the Eddington-Finkelstein coordinates ($u = t - r^*$, $v = t + r^*$) in region $I$ (for which $U < 0$, $V > 0$). A similar prescription for coordinates $\bar{u}, \bar{v}$ can be made in order to describe region $II$ (for which $U > 0$, $V < 0$). Using these coordinate systems and the clock of the boosted observer, ingoing modes depend on $(V, \Omega)$, and outgoing modes on $(U, \Omega)$.

The commutation relations in eqns. (1) and (2) are a result of the gravitational backreaction of modes in the vicinity of the black hole horizon as shown in [15–17]. This algebra is intriguing in that it elucidates the quantum gravitational nature of the effect; on the one hand it becomes exactly zero when $G/R^2 \to 0$, and on the other, it reduces to a classical effect as $\hbar \to 0$ when the commutators are replaced by Poisson brackets. Another important aspect of this algebra is that it indicates an inherent uncertainty in determining the exact lightcone position of propagating modes since $\Delta V \Delta U \geq \hbar$; the spacetime has minimal size causal diamonds for each harmonic.

**Symmetries** In the rest of this section, we drop the spectator indices $l$ and $m$ for simplicity, and study the following system:

$$\hat{H} = \frac{1}{2}\left(\hat{U}\hat{V} + \hat{V}\hat{U}\right) = -i\hbar\left(V\frac{\partial}{\partial V} + \frac{1}{2}\right), \quad [\hat{V}, \hat{U}] = i\hbar. \tag{6}$$

The classical counterpart of this Hamiltonian, $H_{cl} = UV$, has the chaotic property that any two nearby solutions diverge exponentially quickly at late times. On the quantum Hamiltonian (6), symmetry under dilations acting as $U' = \kappa U$ and $V' = V/\kappa$ corresponds to time evolutions of the dynamical system, and the $SL(2, R)$ generators in the $V$-basis

$$\hat{P} = -i\partial_V, \quad \hat{D} = -i\Delta - iV\partial_V, \quad \hat{K} = -i2\Delta V - iV^2\partial_V,$$

form the following algebra

$$[\hat{D}, \hat{P}] = i\hat{P}, \quad [\hat{D}, \hat{K}] = -i\hat{K}, \quad [\hat{P}, \hat{K}] = -2i\hat{D},$$

where $\Delta$ is the conformal weight of the $SL(2, R)$ representation. The energy eigenfunctions $\hat{H}\Psi_E = E\Psi_E$, are also to be thought of as eigenfunctions of the dilation operator, $\hat{D}$, with weight $\Delta = 1/2 - iE/\hbar$. It is convenient to split them in odd and even parts under reflection $V \longleftrightarrow -V$ in order to determine which of the two branches to consider near the origin $V = 0$. Their form is $\Psi_E^{even}(V) = |V|^{-1/2+iE/\hbar}/\sqrt{2\pi\hbar}$, $\Psi_E^{odd}(V) = \text{sign}(V)|V|^{-1/2+iE/\hbar}/\sqrt{2\pi\hbar}$; a similar expansion and split holds in the out-basis. They belong to the unitary principal series representations of $SL(2, R)$. The spectrum is hence doubly degenerate (if no further boundary conditions are imposed). These eigenfunctions correspond to plane waves in Eddington-Finkelstein coordinates (5), as in [17].

We can also define a set of *discrete inversions* acting on the space of coordinates as $\hat{I}_{\pm} V = \pm 1/V$. The minus sign is parity odd, while the plus sign is parity even. The energy eigenfunctions are also eigenfunctions of these discrete conformal inversions, acting on the space of functions as

$$\hat{I}_{\pm}\Psi(V) = \frac{1}{|V|^{2\Delta}}\Psi(\pm 1/V), \tag{7}$$

since $\hat{I}_{\pm}\Psi_E^{even}(V) = \Psi_E^{even}(V)$, $\hat{I}_{\pm}\Psi_E^{odd}(V) = \pm\Psi_E^{odd}(V)$. Therefore, the eigenfunctions are also (quasi)-automorphic forms with respect to the discrete $SL(2, R)$ subgroup generated by conformal inversions.

# 3 Identifying CPT conjugate states

**Global identification in spacetime**   In view of the global Kruskal notion of time $2T = U + V$, the CPT transformation acts on spacetime functions as

$$\Phi^{(CPT)}(U, V, \Omega) \;=\; \Phi^{\dagger}(-U, -V, \Omega^P).$$

Using the following relations for the spherical harmonics

$$Y_{lm}^{*}(\Omega) = (-1)^m Y_{l-m}(\Omega), \quad Y_{lm}(\Omega^P) = (-1)^l Y_{lm}(\Omega),$$

a global identification of CPT conjugate classical partial waves results in

$$\Phi_{lm}(U, V) \;\sim\; (-1)^{l-m} \Phi_{l-m}^{\dagger}(-U, -V). \tag{8}$$

In particular this identification on the classical spacetime background of the black hole given in eqn.(4) holds for all $V$ and $U$, resulting in an antipodal identification on the bifurcation sphere $(V, U) = (0, 0)$. This condition projects out of the spectrum all the even-$l$ modes. This is the global spacetime form of the identification proposed in [16], that results in a single spacetime exterior. On the other hand, it is clear that this form of the identification cannot be valid for the wavefunctions, since $U, V$ are conjugate variables due to eqns. (1) and (2). It is the purpose of the next section to describe the correct implementation of the identification on the wavefunctions that properly takes into account the quantum mechanical commutation relations.

**The local identification in phase space and a chaotic spectrum**   The elementary cell in phase space has area $UV = 2\pi\hbar$. Classically the constant energy trajectories are $UV = \pm E$. In the quantum mechanical phase space, due to the commutation relations of equation (6), a geometric identification between all pairs of points on the $V$-$U$ plane is not possible; points are not sharply defined due to the uncertainty principle. A quantum phase space analogue of the identification must be defined. We should also emphasise that in terms of eigenfunctions of the Dilation operator (6), any such identification must be applied to the wavefunctions describing both the ingoing and outgoing modes, since $U, V$ are phase space conjugate variables. Our guide is the discrete inversions of eqn. (7), which are symmetries of the energy eigenfunctions.

In phase space, remaining consistent with the uncertainty principle, the $\hat{I}_{-}$ inversion (in terms of a general parameter[2] $h$) is enlarged into [24]

$$\hat{T}_1^{\pm}: \quad V' = -\frac{h}{V}, \quad U' = \pm \frac{V^2 U}{h}, \tag{9}$$

$$\hat{T}_2^{\pm}: \quad V' = -\frac{h}{U}, \quad U' = \mp \frac{V U^2}{h}. \tag{10}$$

These are area preserving canonical transformations, since the Jacobian matrix of the transformations has a determinant $\pm 1$; the sign depends on the superscript of the generator in question. They were also proposed in [10, 11] as a possible means of obtaining a discrete spectrum for the Dilation operator; we shall show how this is realised in the present example and the relation to CPT. The $\hat{I}_{+}$ inversion is related to $-\hat{T}_{1,2}^{\pm}$ in phase space, so these generators provide a complete phase space description of both transformations. We also notice that $\hat{T}_1^{-}, \hat{T}_2^{+}$ leave the operators $\pm\hat{D}$ and the evolution Hamiltonian $\pm\hat{H}$ invariant, while $\hat{P}$ and $\hat{K}$ change sign. For fixed $UV = h$, the transformations (9), (10) generate a finite group, the Dihedral group $\mathcal{D}_4$ with eight elements. A geometric property of the Dihedral group $\mathcal{D}_4$, is

---

[2]The choice of name for the parameter is intentional. It will turn out to specify Planckian cells.

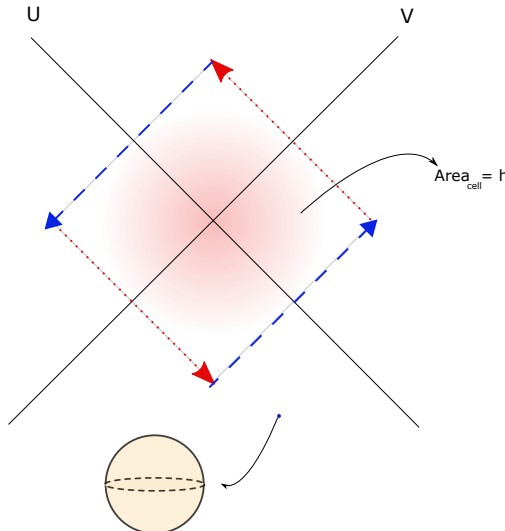

Figure 1: The central causal diamond composed out of four elementary phase space cells. We also depict how the edges are identified under the action of $T_2^+$. There is also an additional transverse $S^2$ at each point.

that it reshuffles the vertices of the square centered at the origin ($U = 0 = V$); we will call this square the central causal diamond. This diamond consists of four elementary phase space cells, each of which has an area $|UV| = h$, so its total area is $4h$. It can also be split into eight fundamental triangular chambers that get interchanged through the action of the elements of $\mathcal{D}_4$. A plot of the central causal diamond and the four cells is provided in fig. 1.

On the other hand, the analogue of the lightcone part of the CPT identification eqn. (8), is imposing an identification under the action of the generator[3] $T_2^+$. This is a legitimate operation, since $\hat{T}_2^+$ belongs in the center of the Dihedral group together with the identity. If we make this identification at the edges of the central causal diamond it endows it with an $\mathbb{RP}_2$ topology.[4] In addition, since $\hat{T}_2^+$ commutes with the Hamiltonian, we can apply a Dilation that transforms the central causal diamond into a distorted parallelogram of sides $L_U, L_V$ (as long as $L_U L_V = h$). This means that the identification has to be performed between all the points of the trajectories $UV = \pm h$, that are mapped into each other under the action of $\hat{T}_2^+$.

At the level of quantum mechanical states, we must impose this identification on the eigenfunctions of the Hamiltonian or Dilation operator. In order to impose the complete CPT identification on the wavefunctions, we reinstate the partial wave indices and expand the wavefunctions in terms of the Dilation eigenfunctions as

$$\Psi_{in}(V,\Omega) = \sum_{lm} \int_{-\infty}^{\infty} dE\, \Psi_{Elm,in}(V)\, Y_{lm}(\Omega),$$

$$\Psi_{Elm,in}(V) = A_{Elm}^{in}\, \Psi_{E,in}(V).$$

A similar expansion holds for the out-wavefunctions. We can again choose to split this expansion using even and odd modes. We begin with the Fourier transform which describes the scattering matrix arising from the gravitational back-reaction, that relates the in/out partial

---

[3]With $h = UV$, the generator $\hat{T}_2^+$ corresponds to the part of CPT that induces $V' = -V$ and $U' = -U$.

[4]A mathematical explanation of this based on the relations between Coxeter elements of the dihedral group, can be found in [24]. Closed paths fall into two homotopy classes depending on the odd or even-ness of the number of reflections along the path. This is because $\pi_1(\mathbb{RP}_2) = Z_2$; therefore, to each path, there is an associated number $\pm 1$.

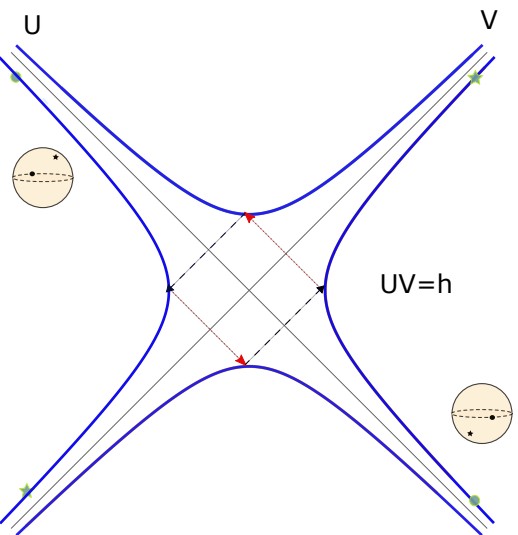

Figure 2: The near horizon CPT identification for the wavefunctions on the locus $h = |UV|$ generated by evolving the central causal diamond of fig. 1 with the Dilation operator. We also depict the complementary identification on the transverse $S^2$.

wave wavefunctions

$$\Psi_{Elm,out}(U) = \int_{-\infty}^{\infty} \frac{dV}{\sqrt{2\pi\hbar}} e^{-\frac{i}{\hbar}UV} \Psi_{Elm,in}(V). \tag{11}$$

Using the explicit form of the ingoing eigenfunctions, the scattering problem can be split into even and odd parts that give a relation between the coefficients

$$A_{Elm}^{out,even} = A_{Elm}^{in,even} (2\hbar)^{+i\frac{E}{\hbar}} \frac{\Gamma\left(\frac{1}{4} + i\frac{E}{2\hbar}\right)}{\Gamma\left(\frac{1}{4} - i\frac{E}{2\hbar}\right)}, \tag{12}$$

$$A_{Elm}^{out,odd} = A_{Elm}^{in,odd} (2\hbar)^{+i\frac{E}{\hbar}} \frac{\Gamma\left(\frac{3}{4} + i\frac{E}{2\hbar}\right)}{\Gamma\left(\frac{3}{4} - i\frac{E}{2\hbar}\right)}. \tag{13}$$

The action of CPT on the complete wavefunctions is then given by the combined action of the operator $\hat{T}_2^+$ in (10), together with complex conjugation and parity on the two sphere so that

$$\Psi_{Elm,out}(U) \rightarrow (-1)^{l-m} \Psi_{El-m,out}^{\dagger}\left(-\frac{VU^2}{h}\right), \tag{14}$$

$$\Psi_{Elm,in}(V) \rightarrow (-1)^{l-m} \Psi_{El-m,in}^{\dagger}\left(-\frac{h}{U}\right), \tag{15}$$

hold for both even and odd modes. The identification of CPT conjugate wavefunctions should then be performed on the "Planckian hyperbolae" $UV = \pm h$ generated from the elementary diamond. This identification is depicted in fig. 2, and replaces the classical identification eqn. (8).

Using the functional relation of the Riemann zeta function (that holds for all the points of the complex plane)

$$\zeta(s) = \pi^{s-1/2} \frac{\Gamma\left(\frac{1-s}{2}\right)}{\Gamma\left(\frac{s}{2}\right)} \zeta(1-s),$$

with $s = 1/2 + iE/\hbar$, and combining the S-matrix relations (12), (13) together with eqns. (14), (15) defining the identification, we obtain the following equivalence relation for the even modes

$$A_{Elm}^{in,even}|V|^{\frac{iE}{\hbar}}\zeta(1/2-iE/\hbar) \;\sim\; A_{El-m}^{out,even\,*}(-1)^{l-m}|U|^{i\frac{E}{\hbar}}\zeta(1/2+iE/\hbar)\,. \tag{16}$$

We should emphasise once more that this equivalence relation holds *for all* the points of the locus $h = |UV|$ (an infinite 1-parametric family of conditions). Due to this it can be satisfied for non-trivial eigenfunctions $A_{Elm}^{in/out,even} \neq 0$, iff $\zeta(1/2-iE/\hbar) = \zeta(1/2+iE/\hbar) = 0$. On the critical line $Re(s) = \frac{1}{2}$, these are precisely the non-trivial Riemann zeros (corresponding to *real* energies $E$).

For the odd waves $\Psi^{odd}$, we have an additional minus sign arising from the $\text{sign}(V)$ in the odd eigenfunctions of eqn. (6), resulting in the equivalence relation

$$A_{Elm}^{in,odd}|V|^{\frac{iE}{\hbar}}\beta(1/2-iE/\hbar) \;\sim\; -A_{El-m}^{out,odd\,*}(-1)^{l-m}|U|^{i\frac{E}{\hbar}}\beta(1/2+iE/\hbar)\,, \tag{17}$$

where we used the functional relation for the Dirichlet $\beta$ function

$$\beta(s) = 2\pi^{s-1/2}\frac{\Gamma\left(1-\frac{s}{2}\right)}{\Gamma\left(\frac{1}{2}+\frac{s}{2}\right)}\beta(1-s)\,.$$

This is a quantisation condition for the spectrum of odd waves $\Psi^{odd}$, that is now given by the zeros of the Dirichlet beta function.

In essence, the quantum boundary condition on the phase space generates a discrete spectrum, providing a regularisation procedure for the Dilation operator that respects the physical symmetries of our quantum Hamiltonian and its associated dynamics. This is a different quantisation of the spectrum compared to the cutoff proposed in [17,25], and takes the phase space structure in eqns. (1) and (2) into account.

The complete spectrum of the Hamiltonian (2), can be decomposed as

$$\frac{E}{\hbar} \;=\; \sum_l \frac{E_l}{\hbar} + \frac{R^2(l^2+l+1)}{8\pi G\hbar} = \frac{4GM}{c^2}\sum_l \omega_l\,,$$

separately for the even and odd modes. $E = E_l^{even}/\hbar$ is given by the zeros of the Riemann zeta function, and $E_l^{odd}/\hbar$ by those of the Dirichlet beta function. The two-fold degeneracy between the even and odd modes is broken. There is still a $(2l+1)$-fold degeneracy due to the $m$ index. The variables $\omega_l$ carry the correct SI units of energy for each partial wave, as registered by the asymptotic observer. This indicates that for even for a black hole of a solar mass, the spectrum is extremely dense, since the low energy gap $\Delta\omega/c^2 \sim \Delta E/\hbar GM_\odot$ is of the order of $M_P/M_\odot \sim 10^{-38}$ in Planck units or $10^{-10}eV/c^2$ for zeroes having an $O(1)$ spacing. A similar decomposition can also be written for the Hamiltonian (1).

In addition to the non-trivial zeros, the conditions given by eqns. (16) and (17), are also satisfied for the trivial zeros

$$\frac{1}{2}\pm i\frac{E_l^{even}}{\hbar} \;=\; -2n\,,\quad \frac{1}{2}\pm i\frac{E_l^{odd}}{\hbar} \;=\; -(2n-1)\,,\quad n\in\mathbb{Z}^+\,.$$

These zeros correspond to the poles (and zeros) of the scattering matrix, defined via eqns. (11), (12), (13), on the axis of negative (and positive) imaginary energy, and thus to unstable resonances.

**The lightcone momentum operators**    We now study the effect of the local phase space identification on the spectrum of the lightcone momentum operators $\hat{P}_V$, $\hat{P}_U$. Their eigenfunctions are Kruskal waves

$$\Phi_{P_V}^{in}(V) = c^{in} e^{iP_V V}, \quad \hat{P}_V \Phi_{P_V}^{in}(V) = P_V \Phi_{P_V}^{in}(V),$$
$$\Phi_{P_U}^{out}(U) = c^{out} e^{iP_U U}, \quad \hat{P}_U \Phi_{P_U}^{out}(U) = P_U \Phi_{P_U}^{out}(U).$$

Reintroducing again the $l, m$ indices, the identification under the action of $\hat{T}_2^+$ (for $UV = h$), results into

$$\Phi_{P\,lm}^{in}(V) \sim (-1)^{l-m} \Phi_{P\,l-m}^{in\dagger}\left(-\frac{h}{U}\right), \quad \text{for} \quad UV = h,$$

leading to $c_{Plm}^{in} = (-1)^{l-m} c_{-Pl-m}^{in*}$, with a similar equation for the out modes. We therefore find that the continuous spectrum of these operators is unaffected by the identification we propose. This is a first indication that the horizon is smooth for an observer travelling along the Kruskal null rays.

## 4    Conclusions

Inspired by the expectation that there are no global symmetries in quantum gravity, we observe that treating CPT as a local gauge symmetry leads to an effective boundary condition, relating points across the two sides of the Penrose diagram that belong to the elementary Planckian hyperbolae[5] $|UV| = h$. This results in a discretisation of the spectrum of the near horizon Dilation operator when appended with extra quantum numbers; in the present construction, these arise from the presence of extra transverse dimensions. We should also mention at this point another possible spectral interpretation of the equivalence relations (16) and (17), as providing for missing spectral lines (absorption spectrum in the continuum), along the lines of work by Connes [27,28]. Whether our conditions describe absorption, emission, or both,[6] requires a careful comparison between the semi-classical trace formulae and the density of states arising from the Riemann zeta function, this is an interesting problem worthy of further study.

The resulting spectrum is extremely rich, arguably chaotic, and supports several properties expected of quantum black holes. It goes to show that a simple, unbounded Hamiltonian may turn out to have an extremely interesting spectrum provided appropriate boundary conditions are chosen.

Should the approach of Berry and Keating [10,11] still provide an interesting way forward, the boundary condition and the Hamiltonian proposed in this article may help the study of the Riemann hypothesis. An asymptotic analysis of a smooth approximation of the density of zeros of the Riemann zeta function may be an interesting starting point to compute the partition function of this model, and possibly derive the Bekenstein-Hawking entropy of the Schwarzschild black hole. This will clarify whether the present proposal adequately captures all the black hole microstates.

A study of multi-particle scattering cross sections and the consequence of this boundary condition on such observables is another future direction of physical interest. The presence of a discrete spectrum that is unbounded from below, remains a puzzle to be understood; perhaps our proposal of gauging CPT is insufficient to fix the ambiguity in the energy of the ground state in this system.

---

[5]These surfaces are reminiscent of the stretched horizon proposal [26].

[6]Physically we would expect the black hole to be able to both absorb and emit at the same frequencies.

# Acknowledgements

It is a pleasure to acknowledge various helpful discussions with E. Kiritsis, K. Papadodimas, M.M. Sheikh-Jabbari, T. Tomaras, N. Tsamis, and E. Verlinde, on topics related to the black hole S-matrix. Special thanks go to Gerard 't Hooft for pointing out to us that the naive cutoff regularisation is in clash with the symmetries of the Dilation operator and the S-matrix, and for several discussions over the years. We thank David Tong for his talk on gauging discrete symmetries at the Amsterdam String Workshop 2019. We also thank Michael Berry and Jon Keating for correspondence and feedback on a draft of this article.

**Funding information**   P.B is supported by the Advanced ERC grant SM-GRAV, No 669288. N.G is supported by the Delta-Institute for Theoretical Physics (D-ITP) that is funded by the Dutch Ministry of Education, Culture and Science (OCW).

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
