# Peer review of "Black holes, quantum chaos, and the Riemann hypothesis"

_SciPost Physics Core, doi:SciPost Phys. Core 4, 032 (2021)_

## Round 3 · Referee Report · Anonymous (Referee 1) · 2020-10-11

Strengths

1) the paper addresses an important problem in quantum gravity 2) it proposes an interesting connection with a seemingly unrelated problem as the Riemann hypothesis 3) the authors have a good knowledge of the literature 4) the paper is generally well written

Weaknesses

1) the paper lacks clarity in the sections that support the main claim

Report

The authors analyze in this paper the quantization of two Hamiltonians that describe the near-horizon dynamics of a Schwarzschild blackhole. The corresponding S-matrices were derived in a previous paper [17] by the same authors showing that they are equal to the one found by ‘t Hooft that describes the gravitational back reaction of the blackhole [15,16]. The Hamiltonians, given in eqs. (1) and (2), are written in terms of the Kruskal variables U and V whose commutator is the Heisenberg one for x and p, up to some constants for the Hamiltonian (2). The U-V variables also carry the orbital quantum numbers l,m thanks to the rotational symmetry of the Schwarzschild metric.

To analyze this model the authors drop the l, m dependence obtaining the Hamiltonian H= (U V+ VU)/2 that Berry and Keating conjectured to be related to Riemann hypothesis (this is the famous xp model) [10,11]. More explicitely, these authors proposed that a certain quantization of H= xp would yield a discrete spectrum given by the non trivial zeros of the Riemann zeta function. To do so, Berry and Keating imposed certain boundary conditions on the wave function but they were not consistent. The aim of the authors in this paper is to find consistent boundary conditions for the UV (i.e. xp) Hamiltonian that would yield the Riemann zeros in the spectrum. Those boundary conditions are derived from the gauging of the CPT symmetry in quantum gravity, that several authors have proposed in the past.

I find very appealing the ideas proposed in this paper, specially the connection between the near-horizon dynamics and the xp Hamiltonian. However, I do not
agree with the conclusion, namely that imposing the CPT symmetries yield boundary conditions that discretize the spectrum that then corresponds to the zeros of the Riemann zeta function or the Dirichlet beta function.

Let me explain this statement in detail. The problem is how the discrete symmetry T^+_2 is implemented. The authors say that it leaves the center (U,V)=(0,0) invariant, but using eq. (10) one obtains (0, infinity) under the transformation. I understand that the goal of the authors is to adapt somehow, in the blackhole geometry, the Berry-Keating idea of the Planck cell, but this is not well explained in the manuscript. I suggest the authors to clarify this part. Let me assume for the time being that there are indeed consistent boundary conditions based on CPT. These conditions lead to eq.(14) that the authors arguee is satisfied provided the energy E is the imaginary part of a Riemann zero. It is not clear that this is the case. First of all, the lhs of this equation depends on V while the rhs depends on U. I guess, one has to impose that |UV| = h and then use equations (12) and (13). If so, please do it explictely. But the main problem here is that for a Riemann zero the lhs and rhs of eq.(14) actually vanish, so is not clear that this is an eigenvalue equation.
Can one write (14) as zeta(1/2 + i E)=0? Something like this also appears in the Berry-Keating paper, where a condition was derived so that the wave function vanishes (see eq.(25) and below of [10}). The vanishing of the lhs and rhs could rather be interpreted, not as bound states, but as missing states along the lines of Connes spectral realization ot the Riemann zeros. If this were the case it would be a very interesting result too.
The paper satisfies the general criteria for publication in this journal, but before acceptance the authors have to answer the questions posed above.

Requested changes

1) After eq.(5) it is said that region II in the Penrose diagram corresponds to U > 0 and V < 0. I think there is V >0.
2) At the top of page 5 it is written:
“Demanding complete conformal invariance on the wavefunctions trivialises them.”
What is meant by that?
3) In page 5 it is given the CPT transformation of the wave function. They should be given both for psi(U) and psi(V) independently since they are conjugate variables.

  • validity: good
  • significance: high
  • originality: high
  • clarity: low
  • formatting: good
  • grammar: excellent

Author:  Nava Gaddam  on 2020-10-12  [id 999]

(in reply to Report 1 on 2020-10-11)
Category:
answer to question

We would like to first thank the referee for the careful reading of our paper, and for the encouraging feedback. We understand the referee's concerns as being twofold:

a) Clarification of the boundary condition. In retrospect, the discussion of the boundary condition in the paper is perhaps unclear, and distracted by the discussion of the Dihedral group. The operator T_2^+ is a symmetry of the Hamiltonian. |UV| = h defines a hyperbolae and any point on such a hyperbola is time translated in these coordinates by dilation. Now, consider the central causal diamond (centered around U=0=V) of area 4h. This diamond is made of four squares of area h (one in each quadrant). One of the four vertices of the central causal diamond lies in Region I (with coordinates U= - \sqrt{h} and V = \sqrt{h}) and another lies in Region II (with coordinates U = \sqrt{h} and V = - \sqrt{h}). The effect of dilations is such that each of the squares making up the central causal diamond is elongated into a rectangle with the same area as the square, namely h.

The boundary condition then identifies hyperbolae passing through these vertices. This is our concrete proposal for how to adapt the boundary condition of Berry-Keating. This ensures that the Planckian uncertainty in going to the ultra local points within Planckian distance of the horizon is accounted for.

Please see the attached images for a picture of the central causal diamond, and the hyperbolae being referred to.

b) The eigenvalue equation. Equation (14) is arrived at, by using s = 1/2 + i E/\hbar, and combining the S-matrix relations (12) and (13) together with the unnumbered equations defining the transformation of the wavefunctions under CPT. Now, equation (14) is an equivalence relation that must hold for all U and V on the hyperbolae |UV| = h (therefore, it is an infinite number of conditions on the hyperbola). Of course, one trivial way to satisfy this equivalence relation is to demand that all fourier coefficients of the wavefunctions be zero. This trivialises the wavefunctions. So, the only way to satisfy this equivalence relation for non-trivial wavefunctions is by independently setting \zeta(1/2+i E/\hbar) = 0 *and* \zeta(1/2 - i E/\hbar) = 0. Therefore, we believe that this is precisely what the referee desires.

--- The authors

P.S: Subject to the Editor's approval, we are happy to clarify these points and also attend to the changes requested by the referee.

Attachment:

images.pdf

---

## Round 4 · Referee Report · Anonymous (Referee 2) · 2021-6-22

Report

Dear Editor,

In this paper, authors demanded a CPT invariance of the wavefunction, that in turn gives certain condition on U and V operators. Then they have considered the generators of Dilatation (the Hamiltonian mentioned in equation (1) ) of the paper. Using these conditions, one can show that after solving the spectrum of the Hamiltonian, the spectrum consists of even and odd modes. Even modes are given by the Riemann zeta function and the odd modes are given by zeroes Dirichlet beta function. Then authors comment that using the idea put forward by Berry and Keating may put forward an interesting playground to study the Riemann hypothesis.

This may be an interesting connection but I do not see its obvious utility. Maybe the authors can elaborate or review the idea of Berry and Keating a bit more and make a concrete connection or at least a bit more pointers on how their analysis will provide me with an interesting insight into the Riemann hypothesis. So far the far merely points out a mathematical relation that may be interesting.

Also, the authors claim in the abstract "It strengthens the
proposal that this quantum Hamiltonian captures the near horizon dynamics of
the scattering matrix of the Schwarzschild black hole, given the rich chaotic spectrum upon discretisation. " Except for some passing comment about the phase space in some places of the paper (eg the last paragraph on page 5) I do not see any concrete calculation which supports the statement mentioned in the abstract. There are other studies of the spectrum which points to the act the there is a rich chaotic structure, are the authors simply referring to those? or have they done some more analysis in this paper? (which is not obvious at least to me).

This paper in its current form may be considered for sci-post physics core if the editor thinks it is appropriate.
  • validity: ok
  • significance: low
  • originality: ok
  • clarity: good
  • formatting: reasonable
  • grammar: good

Author:  Nava Gaddam  on 2021-06-30  [id 1536]

(in reply to Report 1 on 2021-06-22)

We are thankful to the referee for reviewing our article. We would like to respond to the two broad comments they have raised.

  1. Since the work of BK, there has been a long program of attempting to construct similar Hamiltonians that may produce the spectrum of non-trivial Riemann zeros (see ref. 14 of the present article and references therein). Connes has also proposed that these might instead be seen as 'missing states' in the spectrum. BK's original work considered an inverted harmonic oscillator (but has some shortcomings and doesn't yet prove the Riemann hypothesis). In our article, the inverted harmonic oscillator arises from the near-horizon dynamics of 4d Schwarzschild black holes and therefore carries angular degrees of freedom arising from the partial waves. These naturally give rise to additional ingredients to the BK proposal inspired by black hole dynamics. We have clearly not carried out a detailed mathematical study of self-adjointness and associated Maslov indices, etc. Without such further careful study, it is unclear exactly what our proposal means for a proof of the Riemann hypothesis. Nevertheless our paper posits that it is intimately tied to black hole physics.

  2. 't Hooft has long argued that gravitational interactions near the horizon have an important role to play in the information paradox. In our previous work in 2016 (ref. 17 of the current article), we showed that these interactions can be recast in the form of an infinite tower of inverted harmonic oscillators. The natural question is whether such simple inverted harmonic oscillators can capture the rich, chaotic dynamics expected of black holes. In the present article, we have shown the existence of a physically motivated boundary condition on our proposed quantum Hamiltonian that gives rise to a spectrum of non-trivial Riemann zeros (eqs 16 and 17, and the discussion thereafter). This is the main result of the calculation we presented in this article. Therefore, the statement, from the abstract, quoted by the referee is justified by the main calculation we have performed in this article. 

Thank you for your consideration, The authors.

---

## Round 4 · Referee Report · Anonymous (Referee 1) · 2021-8-12

Strengths

1) the paper deals with a very interesting problem connecting several areas on theoretical physics and mathematics 2) the derivation of results is rigorous and can be easily followed 3) it contains interesting suggestions concerning the realization of the so called Riemann zeros with the emission or absortion spectrum of a Hamiltonian 4) The latter Hamiltonian is given by the Berry Keating xp model supplemented by certain boundary conditions

Weaknesses

1) the term "quantum chaos" appears in the title but it is not clear, nor described in some detail, how it may appears in the results presented except for some general considerations

Report

This paper is worth to be published in this journal. It contains very interesting results that may motivate the research in a fundamental problem such as relating black holes and the Riemann hypothesis

Requested changes

The new version answers positively the comments made in my previous report

---

## Round 4 · List of Changes

Dear editor,

As per your suggestions, we have modified the draft to address the recommendations of the referee:

Requested change 1) The definition of region II we use is indeed what we write, it is intended to be the second exterior of the Kruskal diagram. Sometimes the interior is called region II, perhaps leading to the referee's confusion (this interior is U>0, V>0). So no change has been made here. We have also added the figures we used in our response to the referee to the paper, for clarification.

Requested change 2) We have removed the confusing sentence the referee points to; it is no significance to the rest of the paper.

Requested change 3) In equation 8, the identification written is for classical functions (as we now clarified), which indeed does not take the conjugate nature of the variables into account, as the referee points out. The appropriate identifications are what we present in equations (14) and (15) in the new version. As the referee asks, these are indeed written as two equations, one for \psi(U) and another for \psi(V). We have also added references to Connes' work.

We hope that this suffices and that you would accept it for publication. Thank you for your consideration.
The authors.

---

## Editorial Decision

published